# Optimization of cataract surgery follow-up: A standard set of questions can predict unexpected management changes at postoperative week one

**Giannis A. Moustafa**[1], **Durga S. Borkar**[1,2], **Sheila Borboli-Gerogiannis**[1], **Scott H. Greenstein**[1], **Alice C. Lorch**[1], **Ryan A. Vasan**[1], **Carolyn E. Kloek**[1,3]\*

1 Department of Ophthalmology, Massachusetts Eye and Ear, Harvard Medical School, Boston, Massachusetts, United States of America, 2 Retina Service, Wills Eye Hospital, Philadelphia, Pennsylvania, United States of America, 3 Dean McGee Eye Institute, University of Oklahoma College of Medicine, Oklahoma City, Oklahoma, United States of America

\* carolynkloek@gmail.com

**Data Availability Statement:** Raw data underlying the study cannot be made publicly available due to ethical restrictions imposed by the Mass. Eye and

## Abstract

### Purpose

There is limited evidence to inform the optimal follow-up schedule after cataract surgery. This study aims to determine whether a standardized question set can predict unexpected management changes (UMCs) at the postoperative week one (POW1) timepoint.

### Setting

Massachusetts Eye and Ear, Harvard Medical School.

### Design

Prospective cohort study.

### Methods

Two-hundred-and-fifty-four consecutive phacoemulsification cases having attended an examination between postoperative days 5–14. A set of 7 'Yes' or 'No' questions were administered to all participants by a technician at the POW1 visit. Patient answers along with perioperative patient information were recorded and analyzed. Outcomes were the incidence of UMCs at POW1.

### Results

The incidence of UMCs was zero in uneventful cataract cases with unremarkable history and normal postoperative day one exam if no positive answers were given with the question set demonstrating 100% sensitivity (p<0.0001). A test version with 5 questions was equally sensitive in detecting UMCs at POW1 after cataract surgery.

Ear and Partners Healthcare IRB. Upon request, de-identified data will be shared along with any information necessary to interpret the data, such as study protocols, data instruments and survey tools. Prior to that, the detailed plan to execute these methods must be reviewed and approved by the IRB, according to the Partners Healthcare policy. We will ensure long-term data storage and availability by consistently renewing our IRB protocol and by maintaining electronic files of the dataset in more than one encrypted computers/ USB devices and in the Mass. Eye and Ear file hosting service. Researchers interested in the data may contact any of the authors as follows: Giannis Moustafa, Mass. Eye and Ear, Tel: (617) 523-7900, E-mail: giannis_moustafa@meei.harvard.edu; Durga Borkar, Wills Eye Hospital, Tel: (215) 928-3000, E-mail: durga_borkar@meei.harvard.edu; Sheila Borboli-Gerogiannis, Mass. Eye and Ear, Tel: (617) 573-3202, E-mail: sheila_borboli-gerogiannis@meei.harvard.edu; Scott Greenstein, Mass. Eye and Ear, Tel: (617) 573-3202, E-mail: scott_greenstein@meei.harvard.edu; Alice Lorch, Mass. Eye and Ear, Tel: (617) 936-6156, E-mail: alice_lorch@meei.harvard.edu; Ryan Vasan, Mass. Eye and Ear, Tel: (617) 936-6156, E-mail: ryan_vasan@meei.harvard.edu; Carolyn Kloek, Dean McGee Eye Institute, Tel: (405) 271-1090, E-mail: carolyn-kloek@dmei.org.

**Funding:** This study was funded by a research grant from the American Society of Cataract and Refractive Surgeons (https://ascrs.org/), Fairfax, VA, USA (DSB, CEK). The funder had no role in study design, data collection and analysis, decision to publish, or preparation of the manuscript.

**Competing interests:** The authors have declared that no competing interests exist.

## Conclusion

In routine cataract cases with no positive answers to the current set of clinical questions, a POW1 visit is unlikely to result in a management change. This result offers the opportunity for eye care providers to risk-stratify patients who have had cataract surgery and individualize follow-up.

## Introduction

The Centers for Medicare and Medicaid Services (CMS) Data Compendium ranks cataract code 66984 (extracapsular cataract extraction with insertion of intraocular lens) as the single largest expenditure for all Part B procedures with costs estimated at over $2 billion annually, accounting for 1.8% of total allowed charges.[1] The cost of cataract surgery with intraocular lens implantation for a Medicare beneficiary was estimated to be $2335 in 2010 when performed in the Ambulatory Surgery Centers (ASC) setting, with the fee increasing for surgeries performed as part of the Hospital Outpatient Departments (HOPD) program.[2] The surgical procedure accounts for only 14% of the total cost billed to patients and insurance providers, with the largest portion of charges coming from surgical facility fees, medications, and eye exams.

Postoperative visits contribute to this cost for both health care organizations and patients. While these examinations are important check-points to ensure appropriate recovery from surgery and to implement management changes to optimize postoperative visual outcome, these examinations also utilize clinic space and resources including staff and physician time. They also add cost to patients, requiring time and trips to the physician's office for the examination. Optimization of cataract postoperative follow-up to deliver the safest and most efficient care has the potential to increase patient satisfaction and reduce overall cataract surgery cost by decreasing the frequency of postoperative visits in a subset of patients who are low-risk for management change and identifying the subset who need to be followed more closely for early diagnosis of complications.

Most commonly in the United States, patients are seen postoperatively on day 1, one week, and one month after cataract surgery. The 2016 American Academy of Ophthalmology Preferred Practice Pattern for Cataract in the Adult Eye recommends that patients who have undergone cataract surgery have their first postoperative visit within 24–48 hours after surgery. [2] A final manifest refraction for the appropriate eyeglasses prescription is also recommended 1–4 weeks after surgery when the anatomical structures have usually recovered and measurements have been stabilized.[2] Our previous study showed a low rate of unexpected management changes (UMCs) at the postoperative week 1 (POW1) timepoint after cataract surgery in uneventful cases with unremarkable ocular history and normal postoperative day 1 (POD1) exam, suggesting that a POW1 visit may be safely omitted in a subset of patients who underwent routine cataract surgery.[3] Nonetheless, some ophthalmic practitioners may feel hesitant to skip the POW1 visit in all routine cases. Hence, we present an alternative method of cataract surgery follow-up at POW1 which allows identification of patients in need of management change with greater sensitivity. In this study, we investigate the ability of a standardized set of questions administered by an ophthalmic technician at the POW1 timepoint to predict UMCs.

## Methods

In this prospective cohort study, the probability and magnitude of patient harm or discomfort anticipated as a result of responding to the clinical questions was not greater than those

ordinarily encountered during the performance of routine eye visits, i.e., the research presented no more than minimal risk of harm to participants. Moreover, no procedures for which written consent is normally required outside of the research context were involved. Verbal consent was obtained by the ophthalmic technician during the pre-exam work up. Prior to checking vision or performing any elements of the exam, the technician asked the patient if he/she was willing to answer a standard set of questions about the operative eye. Patients were told that these questions were directly related to their post-operative care, represented typical questions asked of patients related to post-operative care in a standardized manner, and would take less than one minute to answer. Patients were also told that, after the questions were asked and answered, they could offer additional information to the technician regarding the state of the eye that was recorded in the medical record. The technician then completed the pre-exam work up including checking vision, pupils, motility, and eye pressure prior to the patient's assessment by the physician. Patients were given the option to defer answering the standard set of questions; in which case their pre-exam work up would proceed in a standard fashion. The study and the aforementioned protocol were approved by the Massachusetts Eye and Ear Infirmary Institutional Review Board. All work was compliant with the Health Insurance Portability and Accountability Act and the manuscript was structured in accordance with the Standards for the Reporting of Diagnostic Accuracy Studies (STARD) statement.[4]

## Question set

An electronic survey was administered to the 16 cataract surgeons in the Comprehensive Ophthalmology Service at Mass. Eye and Ear. Faculty were queried about UMCs that occur at the POW1 visit and the etiologies that lead these management changes. Based on these responses, a set of questions was then created that linked each potential UMC identified to a symptom or other cause that could be expressed by patients as a positive answer to a question (Table 1). The questions were created with the goal of being simple, brief, and understandable to

**Table 1. Questions asked during the postoperative week 1 visit.** Abbreviations: AC anterior chamber; IOL intraocular lens; IOP intraocular pressure; UMC unexpected management change.

| Question | Answer | | Etiology[potential UMCs] |
|---|---|---|---|
| 1. Are you having eye pain? | Yes | No | **Endophthalmitis[a,b]; Epithelial defect[a,b]; AC inflammation[a,b,d]; Retained lens fragment[c,d]; Elevated IOP[a,b,c,d]** |
| 2. Are you having increasing eye redness? | Yes | No | **Endophthalmitis[a,b]; AC inflammation[a,b,d]; Retained lens fragment[c,d]; Elevated IOP[a,b,c,d]** |
| 3. Are you unhappy with your vision? | Yes | No | **Endophthalmitis[a,b]; Corneal edema[b]; IOL out of position[c,d]; Epithelial defect[a,b]; AC inflammation[a,b,d]; Vitreoretinal pathology[b,c,d]; Retained lens fragment[c,d]** |
| 4. Has your vision decreased since your last visit? | Yes | No | **Endophthalmitis[a,b]; Corneal edema[b]; IOL out of position[c,d]; Epithelial defect[a,b]; AC inflammation[a,b,d]; Vitreoretinal pathology[b,c,d]; Retained lens fragment[c,d]** |
| 5. Do you have an increase in floaters? | Yes | No | **Vitreoretinal pathology[b,c,d]; Endophthalmitis[a,b]** |
| 6. Do you have new flashing lights? | Yes | No | **Vitreoretinal pathology[b,c,d]** |
| 7. Do you understand your eye drops? | Yes | No | **Noncompliance with postoperative care[a,b,c,d]** |

[a]Deviation from the eye drop taper plan prescribed at postoperative day one for the antibiotic, steroid, or nonsteroidal anti-inflammatory drops

[b]Addition of an eye drop excluding artificial tears

[c]Performance of a procedure excluding suture removal

[d]Urgent or emergent referral to a specialty ophthalmology service

patients; an effort was made to create the shortest total number of questions possible that would still permit screening for the full set of possible management changes that may symptomatically present at POW1.

## Study population and case selection

The study population comprised of the patients of five cataract surgeons at Mass. Eye and Ear who underwent surgery between April 3, 2017, and October 19, 2017, returned for a visit between the postoperative day 5 and postoperative day 14 timeframe, and provided verbal informed consent. Patients with preoperative characteristics that would necessitate a POW1 visit, complicated cases, or patients with any unexpected findings on POD1 exam warranting a POW1 visit were excluded (S1 Fig), as described below:

Preoperative characteristics that would necessitate a POW1 visit included a steroid intraocular pressure response or rebound iritis in patients who had already had cataract surgery in the fellow eye.

Surgery was considered complicated if any of the following intraoperative events was documented: (1) posterior capsule tear, (2) anterior capsule rent, (3) anterior vitrectomy, (4) zonular dehiscence, (5) placement of a capsular tension ring, (6) placement of an intraocular lens in the sulcus or anterior chamber, (7) nuclear fragments dropped in the vitreous, or (8) performance of a concurrent vitreoretinal procedure.

POD1 exclusion criteria were the following: (1) intraocular pressure (IOP) greater than or equal to 30 mmHg in the operative eye in patients without a noted history of glaucoma, ocular hypertension, or glaucoma suspect, (2) IOP greater than or equal to 21 mmHg in the operative eye in patients with a noted history of glaucoma, ocular hypertension, or glaucoma suspect, (3) a wound leak, including trace-positive Seidel test, (4) epithelial defects, including punctuate epithelial erosions and epithelial defects at the incision or paracentesis sites, (5) retained lens fragment, (6) intraocular lens out of position, (7) clinically significant corneal edema, (8) performance of an anterior chamber paracentesis, (9) additional IOP-lowering drops prescribed other than drops used preoperatively, and (10) adjustment of the frequency of either the steroid, antibiotic, or nonsteroidal anti-inflammatory (NSAID) drops compared to the surgeon's standard regimen.

Finally, cases were excluded if they underwent a procedure between the POD1 and POW1 visit (S1 Fig).

Sample size was determined based on preliminary outcomes in order to obtain a statistical power of 80%.

## Intervention and data collection

The structured set of questions was administered to patients at the POW1 examination (Table 1). Patients were asked the questions by an ophthalmic technician prior to performing any other element of the history or examination and prior to any interaction with the physician. Patients were instructed to provide a 'yes' or 'no' response to each question and one-word answers were recorded. Presence of eye pain, increasing eye redness, unhappiness with vision, decreased vision, increased floaters, new flashing lights, and not understanding eye drops were defined as positive answers. Information from the preoperative consultation closest to the date of surgery, operative report, POD1 visit, and POW1 visit were also recorded for each case.

## Outcome

The primary outcome of this study was an UMC at POW1. An UMC was defined as (1) a deviation from the eye drop taper plan prescribed at POD1 for the antibiotic, steroid, or NSAID

drops, (2) addition of an eye drop excluding artificial tears, (3) performance of a procedure excluding suture removal, or (4) urgent or emergent referral to a specialty ophthalmology service.

## Statistical analysis

The Stata version 15 (StataCorp LP, College Station, TX, USA) and Microsoft Excel version 2016 (Microsoft Corporation, Redmond, WA, USA) were used for data analysis. Continuous variables were expressed as means and standard deviations. Categorical variables were summarized using frequencies and percentages. Differences between categorical variables were evaluated using the Fisher exact test. Receiver operating characteristic (ROC) analysis was performed to evaluate the optimal number of questions that must be incorporated into the screening tool and the optimal cutoff value of positive answers. Logistic regression was used to test the association between ordinal variables with unordered dichotomous variables. The alpha level of statistical significance was set at 0.05 and all p values were two-tailed.

## Results

Out of the 254 consecutive cataract cases, 170 (66.9%) uneventful cases with unremarkable history and normal POD1 exam were included in the analysis. Baseline demographic characteristics are described in Table 2.

The incidence of UMCs at POW1 was 8 (4.7%) cases. The most common UMC was a modification of the taper plan prescribed at POD1 for the antibiotic, steroid, or NSAID drops in 6 (3.5%) cases, followed by the addition of an additional eye drop, such as IOP-lowering, antibiotic, or NSAID drops, in 2 (1.2%) cases, and referral to a specialty ophthalmology service in 2 (1.2%) cases. No procedure other than suture removal was performed during POW1 visit (S1 Table).

Of the 50 patients with at least one positive answer, 8 (16%) experienced an UMC. The incidence of UMCs in the 120 patients without positive answers was zero (p<.0001). Responses which were significantly associated with UMCs were reporting of eye pain (p = .018), redness (p = .002), unhappiness with vision (p<.0001), vision decrease (p<.0001), and no understanding of the prescribed eye drops (p = .005). Reporting of new flashes or increased floaters one week after cataract surgery were not associated with UMC (Table 3).

The incidence of UMCs tended to increase as the number of positive answers reported by each patient case increased (p<.0001, logistic regression; S2 Fig).

In order to determine the minimum number of questions required to achieve a high detection rate of UMCs, the screening properties of two question sets, one consisting of the full series of 7 questions (Test 1) and one of only questions with high predictive capacity as determined by Fisher exact test (Test 2), were compared using ROC analysis (Fig 1). In Test 2, the questions querying about the presence of flashes and floaters were excluded. Both questionnaires demonstrated excellent discriminative ability with the AUC being 0.92 (95% confidence interval, 0.87–0.98) for Test 1 and 0.95 (95% confidence interval, 0.92–0.99) for Test 2. Sensitivity did not differ across all cutoff points, and at the cutoff of at least one positive answer both Tests displayed 100% sensitivity in detecting patients in need for UMCs (p<.0001). Test 2 was more specific (84.6% versus 74.1%), and since it is the simplest version of the question set, it may be more appropriate for the initial screening of patients at POW1.

## Discussion

In this study, absence of positive answers to a standardized set of questions at the POW1 timepoint after cataract surgery predicted no UMCs with 100% sensitivity in uneventful cases with

**Table 2. Baseline demographic characteristics of cataract cases.** Abbreviations: OD right eye; OS left eye; POD1 postoperative day 1; SD standard deviation.

|  | Mean±SD or No. (%) |
|---|---|
| **Age, years** | 69.47±10.37 |
| **Gender** | |
| **Male** | 78 (45.9) |
| **Female** | 92 (54.1) |
| **Race/Ethnicity** | |
| **White** | 103 (60.6) |
| **Black/African American** | 25 (14.7) |
| **Asian** | 7 (4.1) |
| **Hispanic/Latino** | 10 (5.9) |
| **Other** | 10 (5.9) |
| **Not available/Declined to declare** | 15 (8.8) |
| **Operative eye** | |
| **OD** | 84 (49.4) |
| **OS** | 86 (50.6) |
| **Second Eye** | |
| **Yes** | 65 (38.2) |
| **No** | 105 (61.8) |
|  | Mean±SD or No. (%) |
| **Age, years** | 69.47±10.37 |
| **Gender** | |
| **Male** | 78 (45.9) |
| **Female** | 92 (54.1) |
| **Race/Ethnicity** | |
| **White** | 103 (60.6) |
| **Black/African American** | 25 (14.7) |
| **Asian** | 7 (4.1) |
| **Hispanic/Latino** | 10 (5.9) |
| **Other** | 10 (5.9) |
| **Not available/Declined to declare** | 15 (8.8) |
| **Operative eye** | |
| **OD** | 84 (49.4) |
| **OS** | 86 (50.6) |
| **Second eye** | |
| **Yes** | 65 (38.2) |
| **No** | 105 (61.8) |

unremarkable history and normal POD1 exam. This standard set of questions has the potential to serve as a screening tool either through a phone call or social media communication to individualize cataract surgery follow-up and eliminate the POW1 visit for appropriate patients.

Few studies have examined the outcomes and utility of the POW1 visit after cataract surgery.[3, 5] McKellar and Elder observed a 4.1% incidence of postoperative complications at the 1-week timepoint in a general population of cataract patients, half of which were unexpected. [5] However, 38% of surgeries in this study were performed by trainees, and cataract surgeries performed by both phacoemulsification and extracapsular cataract extraction were included. Moreover, surgeries of patients included in this study were performed between 1996 and 1998 when complications after cataract surgery were more frequent.[6] In our previous study, the

**Table 3. Incidence of unexpected management changes at postoperative week 1 based on patients' responses to the questions.**

| | No. (%) | p value |
|---|---|---|
| **Pain** | | .018* |
| **Yes** | 2 (40.0) | |
| **No** | 6 (3.6) | |
| **Redness** | | .002* |
| **Yes** | 3 (42.9) | |
| **No** | 5 (3.1) | |
| **Unhappy with vision** | | <.0001* |
| **Yes** | 5 (27.8) | |
| **No** | 3 (2.0) | |
| **Decrease in vision** | | <.0001* |
| **Yes** | 4 (40.0) | |
| **No** | 3 (1.9) | |
| **Floaters** | | >.99 |
| **Yes** | 0 (0.0) | |
| **No** | 7 (4.6) | |
| **Flashes** | | >.99 |
| **Yes** | 0 (0.0) | |
| **No** | 8 (5.0) | |
| **Understanding of drops** | | .005* |
| **Yes** | 5 (3.1) | |
| **No** | 3 (33.3) | |
| **7-question set** | | <.0001* |
| **≥1 positive answers** | 8 (16.0) | |
| **No positive answers** | 0 (0.0) | |
| | No. (%) | p value |
| **Pain** | | .018* |
| **Yes** | 2 (40.0) | |
| **No** | 6 (3.6) | |
| **Redness** | | .002* |
| **Yes** | 3 (42.9) | |
| **No** | 5 (3.1) | |
| **Unhappy with vision** | | <.0001* |
| **Yes** | 5 (27.8) | |
| **No** | 3 (2.0) | |
| **Decrease in vision** | | <.0001* |
| **Yes** | 4 (40.0) | |
| **No** | 3 (1.9) | |
| **Floaters** | | >.99 |
| **Yes** | 0 (0.0) | |
| **No** | 7 (4.6) | |
| **Flashes** | | >.99 |
| **Yes** | 0 (0.0) | |
| **No** | 8 (5.0) | |
| **Understanding of drops** | | .005* |
| **Yes** | 5 (3.1) | |
| **No** | 3 (33.3) | |

(*Continued*)

**Table 3.** (Continued)

| 7-question set | | <.0001* |
|---|---|---|
| ≥1 positive answers | 8 (16.0) | |
| No positive answers | 0 (0.0) | |

*Statistically significant

rate of UMCs at POW1 in cataract cases similar to those in the current study was 0.9%, suggesting that surgeons could consider eliminating the POW1 visit in the appropriate subgroup of patients.[3] Alternatively and according to the surgeon's judgement, the current set of questions can be delivered either by phone or social media, introducing a hybrid model of follow-up. Also, since POW1 is commonly the timepoint at which important modifications to the eyedrop regimen are made, such as discontinuing the antibiotic drop and beginning a steroid drop taper, this communication can both ensure compliance with these changes and address any questions or concerns the patient may have.[7]

In this analysis, reporting of pain, redness, dissatisfaction with vision, vision decrease, and lack of understanding of postoperative eye drops one week after surgery were individually associated with a higher incidence of UMCs at that time point. A 'No' response to the question "Do you understand your eye drops?" likely identifies patients who did not understand their

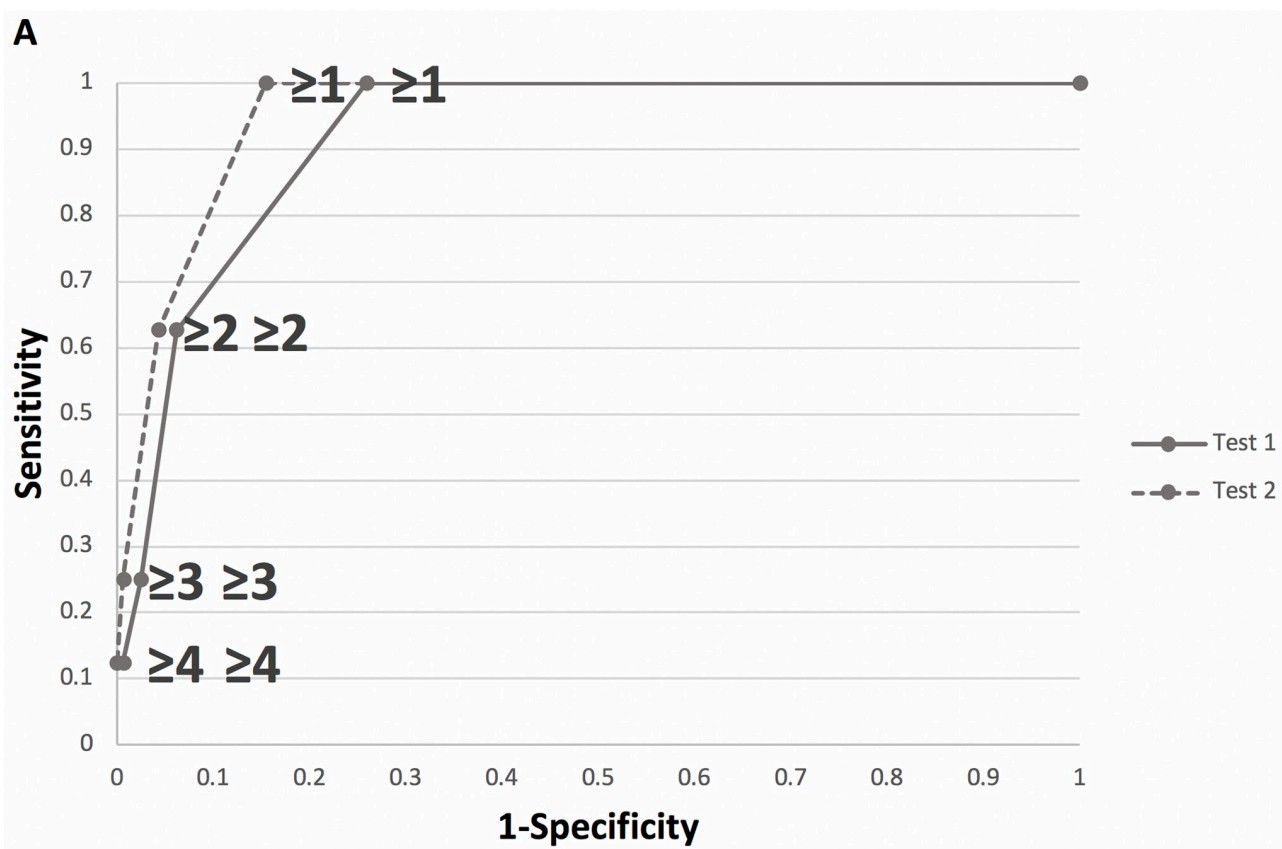

**Fig 1. Receiver operating characteristic (ROC) curve of Test 1 and Test 2 in the screening of unexpected management changes at postoperative week 1 based on the number of positive answers in uneventful cataract cases with unremarkable history and normal postoperative day one exam.**

eye drop regimen or who administer drops incorrectly. A study found that 90% of patients who have not regularly used eye drops before cataract surgery utilize an improper instillation technique.[8] Nonadherence with the postoperative drop regimen is associated with poor outcomes and increased risk for postoperative complications, which likely explains the increased incidence of UMCs in these patients.[8–11]

Patients who indicated they had experienced new flashes and an increase in floaters were not likely to experience UMCs at week one. Retinal complications are rare after cataract surgery, and both flashes and floaters are frequently present in cases with less urgent postoperative events, such as vitreous detachment.[12–17] Moreover, restoration of vision following the removal of the cloudy lens can result in patients reporting an increase of preexisting floaters in the visual field, while dysphotopsias created by the intraocular lens can sometimes be reported as flashes.[18]

The most common unanticipated exam finding at the POW1 visit was an unexpected asymptomatic elevation in the IOP.[3] However, unpublished data from 1931 cases in the Perioperative Care for Intraocular Lens (PCIOL) Study, a retrospective database with perioperative data from cataract surgeries performed at Mass. Eye and Ear, indicated that only 0.2% of uneventful cases with insignificant history and normal exam on POD1 had an IOP of 30 mmHg or more at the POW1 timepoint. Many of these cases were considered to be due to an early steroid response and IOP was expected to normalize without additional treatment with tapering of the steroid drop as scheduled over the course of a few weeks. Also, short-term moderate IOP elevations in otherwise healthy eyes may be permissible after cataract surgery. In the Ocular Hypertension Treatment Study, only one patient out of 819 with elevated IOP between 24 and 32 mmHg developed primary open angle glaucoma at six months follow-up. [19]

Although there were no cases of acute postoperative endophthalmitis in this cohort, endophthalmitis is a "can't miss" complication in the postoperative cataract patient that commonly presents in the first week following cataract surgery. In a study enrolling over 2 million Medicare beneficiaries having undergone cataract surgery, the endophthalmitis rate was 0.63–1.27 cases per 1000 surgeries.[20] In the Endophthalmitis Vitrectomy Study, 98.8% of patients had at least one presenting symptom of either red eye, pain, blurred vision, or swollen lid.[21] Given the low overall frequency of endophthalmitis and the high likelihood that patients with endophthalmitis will experience symptoms, our expectation is that the question set would yield positive answers in patients with endophthalmitis.

Our study has several strengths. Patients were recruited prospectively, and questions were administered prior to the patient-physician encounter, lessening bias of the patients' answers or the technicians' expectations. Questions were kept short and simple and asked in a standardized fashion, in order to make the questionnaire practical and easy to apply in daily practice. Moreover, other than defining the interval of a POW1 visit (days 5–14), other restrictions for patient recruitment were not applied, in order to collect data that represent the "real world". Notably, alternative follow-up schedules eliminating the early follow-up and suggesting a first postoperative examination at 1–2 weeks after surgery or no examination at all have been investigated.[22–29] Although, these schedules may be convenient for clinics in isolated rural areas or patients for whom access to the hospital is difficult,[22, 23, 26] our study was based on the recommendations of the American Academy of Ophthalmology suggesting that a first postoperative evaluation be performed within 24–48 hours from surgery.[2]

This study has limitations. Although the sample size in this analysis is one of the largest among similar studies in the literature and statistical power was estimated to be adequate to detect differences in the incidence of UMCs based on preliminary outcome data, statistical power may be lower than 80% for outcomes with smaller effect size between the groups and

numbers may not be enough to capture the incidence of UMC in patients without positive answers. Moreover, approximately one third of reviewed patients were excluded; however, it was important to set multiple strict exclusion criteria in order to retain only routine cases in the analysis. In addition, although our methodologic design contributes to avoiding several types of bias as described earlier, the possibility of observer-expectancy bias introduced by the fact that attendings were not blinded to the patient responses, which were recorded in the chart, cannot be ruled out. Lastly, the fact that this study was conducted at a single academic institution may limit its external validity.

Optimization of cataract postoperative follow-up is expected to enhance the efficiency of eye care and reduce unnecessary expenditures of patient and physician resources. Implementation of the standard question set outlined above offers eye care providers the opportunity to eliminate POW1 follow-up visits in which a management change is unlikely. Additional benefits include creating additional capacity for more patients to be seen and shorter transportation times and cost for patients. As a next step, a randomized trial accompanied by an analysis of the resulting cost balances will help determine whether this method or universal POW1 omission create value from the perspective of the healthcare system, the physician, and the patient. [3]

In conclusion, a structured set of clinical questions was found to predict the incidence of UMC at POW1 visit after cataract surgery in routine cases. The incidence of UMC in cases with no positive answers is zero, and therefore POW1 check-ups are likely not needed in these patients. This screening test offers the opportunity for case risk stratification and may allow clinicians to substitute POW1 visit with a virtual encounter in the appropriate subgroup of patients. Future studies with larger sample sizes are needed to confirm these findings and further investigate complications such as elevated IOP.

## Supporting information

**S1 Fig. Flow diagram illustrating criteria and number of eyes included and excluded from the study.** For each timepoint, cases are listed under each subcategory for which an exclusion criterion was met (i.e. some cases may be listed under multiple exclusion criteria). Abbreviations: IOL, intraocular lens; IOP, intraocular pressure; NSAID, nonsteroidal anti-inflammatory drug.
(PNG)

**S2 Fig. Incidence of unexpected management changes at postoperative week 1 based on the number of patients' positive answers in the whole set of the 7 questions.**
(TIFF)

**S1 Table. Incidence of unexpected management changes at postoperative week 1.**
(DOCX)

**S2 Table. Data and methods used to reach the conclusions drawn in the manuscript.**
(DOCX)

## Author Contributions

**Conceptualization:** Durga S. Borkar, Carolyn E. Kloek.

**Data curation:** Giannis A. Moustafa, Durga S. Borkar, Carolyn E. Kloek.

**Formal analysis:** Giannis A. Moustafa.

**Funding acquisition:** Durga S. Borkar, Carolyn E. Kloek.

**Investigation:** Giannis A. Moustafa.

**Methodology:** Giannis A. Moustafa.

**Project administration:** Carolyn E. Kloek.

**Resources:** Sheila Borboli-Gerogiannis, Scott H. Greenstein, Alice C. Lorch, Ryan A. Vasan, Carolyn E. Kloek.

**Supervision:** Carolyn E. Kloek.

**Validation:** Giannis A. Moustafa.

**Writing – original draft:** Giannis A. Moustafa.

**Writing – review & editing:** Durga S. Borkar, Sheila Borboli-Gerogiannis, Scott H. Greenstein, Alice C. Lorch, Ryan A. Vasan, Carolyn E. Kloek.

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
