## [Decision Letter · Decision Letter 0]

12 Jul 2019

PONE-D-19-16669

Optimization of Cataract Surgery Follow-up: A Standard Set of Questions Can Predict Unexpected Management Changes at Postoperative Week One

PLOS ONE

Dear Dr Kloek,

Thank you for submitting your manuscript to PLOS ONE. After careful consideration, we feel that it has merit but does not fully meet PLOS ONE’s publication criteria as it currently stands. Therefore, we invite you to submit a revised version of the manuscript that addresses the points raised during the review process.

Please revise your manuscript carefully and answer every question raised by the reviewers. In particular the high problems encountered in day one post op.

We would appreciate receiving your revised manuscript by Aug 26 2019 11:59PM. To enhance the reproducibility of your results, we recommend that if applicable you deposit your laboratory protocols in protocols.io, where a protocol can be assigned its own identifier (DOI) such that it can be cited independently in the future. For instructions see: http://journals.plos.org/plosone/s/submission-guidelines#loc-laboratory-protocols

We look forward to receiving your revised manuscript.

Kind regards,

Cesario Bianchi

Academic Editor

PLOS ONE

Journal Requirements:

2. Please provide additional details regarding participant consent. In the Methods section, please state why it was not possible to obtain written consent, how verbal consent was recorded and whether the ethics committee approved this consent procedure. If your study included minors, state whether you obtained consent from parents or guardians.

Additional Editor Comments:

Dear Dr. Kloek,

Thank you more submitting you interesting work. I received the comments of 3 experts that found the work useful and valid. However, there are comments to be addressed before I can try to reach a final decision.

Please carefully revise your manuscript accordingly to the reviewers comments and answer to each question raised.

I would like that you specifically address comments raised by reviewer#1 regarding the exceptionally high chance of identifying problems in day 1 post op.

Reviewers' comments:

Reviewer's Responses to Questions

**Comments to the Author**

1. Is the manuscript technically sound, and do the data support the conclusions?

Reviewer #1: Yes

Reviewer #2: Yes

Reviewer #3: Yes

2. Has the statistical analysis been performed appropriately and rigorously? 

Reviewer #1: Yes

Reviewer #2: Yes

Reviewer #3: Yes

3. Have the authors made all data underlying the findings in their manuscript fully available?

Reviewer #1: Yes

Reviewer #2: Yes

Reviewer #3: Yes

4. Is the manuscript presented in an intelligible fashion and written in standard English?

Reviewer #1: Yes

Reviewer #2: Yes

Reviewer #3: Yes

5. Review Comments to the Author

Reviewer #1: This paper attempts to provide evidence to change clinical practice in the USA - that being the necessity of performing routine week 1 post op check up following uneventful cataract surgery, when no issues have been identified at day 1 post op. They recommend a questionnaire in which a technician could administer over the phone to reduce unnecessary appointments at which management was not seen to change.

The intention of the work is good, and it is a reasonable attempt to objectively direct post op follow up phone consultations. The questions are simple and clear, and the impact on the study population seems to be valid.

Few comments: in this study population with these 5 surgeons at this institution, there was a 1 in 3 chance of a problem identified at the day 1 post op review – this seems exceptionally high (and was the reason for exclusion in the study). This should be explained in more detail.

It should be unsurprising that due to modern techniques, cataract surgery in routine cases should not have significant issues in the post operative period, and so a validated questionnaire should be able to identify any deviation from this expectation (and therefore pick up cases who may have CMO or endophthalmitis).

The paper is a useful addition to the American literature, but in other parts of the world cataract patients with uneventful surgery and no co-morbidity concerns are not seen at day 1, week 1 and in some instances are only reviewed by their optometrist 6 weeks later. While I am not advocating the pros and cons of this, one could argue there is no evidence base for or against these arbitrary time points aside from the fear of CMO or endophthalmitis. In light of this, these authors should be commended for providing an evidence base to change or direct clinical practice. Their questionnaire may be the first steps in rationalising this practice of review schedules.

However, in many other countries which are not so closely linked to insurance based systems as the USA, the ship has already sailed.

This article could have the message "questionnaire means no more week 1 check ups!") so is a valid cost saving message, if the authors could specifically calculate how using this survey impacted on costs to the patient.

Reviewer #2: This is a well packaged message highlighting the optimal follow-up schedule after cataract surgery, nice reading prior work and how this fits in their overall work.

Reviewer #3: This is a very well written article that questions the current recommended followup schedule after uncomplicated cataract surgery. The study suggests a useful and simple questionnaire to be conducted via virtual visit which can help eliminate postoperative week 1 visit. The sensitivity of the study was excellent at 100%. What was the specificity of the study?

Line 70: The sentence should read as “ Most commonly in the United states, the patients are seen postoperatively on day 1, one week, and one month after cataract surgery.

Line 74: Can you please be specific about couple of weeks? It can be 4-6 weeks after surgery.

Methods: It is important to mention that this is a prospective cohort study here. Also the informed consent should be discussed at the beginning of methods rather than at the end of methods section.

Line 111: The sentence can be re written as “The study population comprised of patients ….informed consent.”.

Line 118: Figure 1 can be moved to results section. Also it is important to explain ROC figure since it is not very self explanatory. It is not clear what the numbers stand for.

Please write figure legends for all the figures at the end of the article

6. PLOS authors have the option to publish the peer review history of their article (what does this mean?). If published, this will include your full peer review and any attached files.

Reviewer #1: No

Reviewer #2: No

Reviewer #3: No

---

## [Author Response · Author response to Decision Letter 0]

30 Jul 2019

We would like to express our appreciation for the thorough and constructive comments provided by the editor and reviewers of this manuscript. In this rebuttal, a point-by-point response is included to each one of these comments. Responses are presented in blue, and the relevant revised text appearing in the manuscript is highlighted using “track changes”.

“Journal Requirements:

http://www.journals.plos.org/plosone/s/file?id=wjVg/PLOSOne_formatting_sample_main_body.pdf and http://www.journals.plos.org/plosone/s/file?id=ba62/PLOSOne_formatting_sample_title_authors_affiliations.pdf”

Confirmed.

“2. Please provide additional details regarding participant consent. In the Methods section, please state why it was not possible to obtain written consent, how verbal consent was recorded and whether the ethics committee approved this consent procedure. If your study included minors, state whether you obtained consent from parents or guardians.”

Thank you for bringing this up. Our study did not include minors. We added the following statement in the beginning of the Methods section (page 5, lines 91-103):

“In this prospective cohort study, the probability and magnitude of patient harm or discomfort anticipated as a result of responding to the clinical questions was not greater than those ordinarily encountered during the performance of routine eye visits, i.e., the research presented no more than minimal risk of harm to participants. Moreover, no procedures for which written consent is normally required outside of the research context were involved. The study and the aforementioned protocol were approved by the Massachusetts Eye and Ear Infirmary Institutional Review Board. All work was compliant with the Health Insurance Portability and Accountability Act and the manuscript was structured in accordance with the Standards for the Reporting of Diagnostic Accuracy Studies (STARD) statement.”

“Additional Editor Comments:

Dear Dr. Kloek,

Thank you more submitting you interesting work. I received the comments of 3 experts that found the work useful and valid. However, there are comments to be addressed before I can try to reach a final decision.

Please carefully revise your manuscript accordingly to the reviewers comments and answer to each question raised.

I would like that you specifically address comments raised by reviewer#1 regarding the exceptionally high chance of identifying problems in day 1 post op.”

Thank you for considering this manuscript. Responses to the reviewers’ comments are presented below.

“Reviewer #1: This paper attempts to provide evidence to change clinical practice in the USA - that being the necessity of performing routine week 1 post op check up following uneventful cataract surgery, when no issues have been identified at day 1 post op. They recommend a questionnaire in which a technician could administer over the phone to reduce unnecessary appointments at which management was not seen to change.

The intention of the work is good, and it is a reasonable attempt to objectively direct post op follow up phone consultations. The questions are simple and clear, and the impact on the study population seems to be valid.

Few comments: in this study population with these 5 surgeons at this institution, there was a 1 in 3 chance of a problem identified at the day 1 post op review – this seems exceptionally high (and was the reason for exclusion in the study). This should be explained in more detail.”

Thank you for accurately summarizing the aims of our study and interpreting our results. Indeed, one third of the total number of screened patients in our study were excluded. Sixty-eight out of the 254 patients (26.8%) were excluded due to POD1 findings. However, 20 of these cases were excluded due to common, minor POD1 issues, such as the adjustment or deviation from the routine antibiotic, steroid, NSAID, or pressure-lowering eye drop regimen. The most common reason for exclusion was elevated IOP, the rate of which in our study is in accordance with the rate reported by other groups. Elfersy et al. found that 14.6% and 4.7% of patients have IOP >23 and >30, respectively, on POD1 after phaco cataract extraction (Elfersy et al., J Glaucoma. 2016 Oct;25(10):802-806). In our study, these numbers were 10.2% (26/254) and 3.9% (10/254), respectively. In our study, the incidence of retained lens fragment on POD1 was 3/254 (1.2%) and the incidence of IOL displacement was 1/254 (0.4%), which are consistent with the study by Tseng et al. investigating intra- and post-operative cataract surgery complications, who found these rates to be 1.7-1.8% and 0.7-1.1%, respectively (Tseng et al., Ophthalmology. 2011 Jul;118(7):1229-35). Regarding epithelial defects, we also excluded minor epi defects, such as punctuate erosions and epi defects at the incision or paracentesis sites (10 out of the 16 cases). For wound leaks, 7 out of the 9 cases demonstrated trace Seidel test, and in terms of corneal edema, we also excluded 2 cases of diffuse and 2 cases of central edema, which were not necessarily severe. In order to make that clearer in the text, we replaced the word “severe” with the word “clinically significant” corneal edema. 

Overall, we chose to exclude even cases with minor POD1 findings, in order to make our final cohort as clear as possible and exclude any case that would potentially warrant a visit at POW1. We made modifications in the text, in order to better explain our POD1 exclusion criteria, as you suggested (page 9, lines 154-157).

“It should be unsurprising that due to modern techniques, cataract surgery in routine cases should not have significant issues in the post operative period, and so a validated questionnaire should be able to identify any deviation from this expectation (and therefore pick up cases who may have CMO or endophthalmitis).”

Thank you for your comment. We agree that most cataract surgery cases do not face any issues in the postoperative period. In our study, our primary outcome was management changes, rather than complications at POW1, which is a direct indicator of the need for a postoperative visit. Although macular edema (Borkar et al., J Cataract Refract Surg. 2018 Jun;44(6):780-781) and endophthalmitis (Jabbarvand et al., Ophthalmology. 2016 Feb;123(2):295-301) are rare complications, these conditions usually present with signs and symptoms, as mentioned in the Discussion.

“The paper is a useful addition to the American literature, but in other parts of the world cataract patients with uneventful surgery and no co-morbidity concerns are not seen at day 1, week 1 and in some instances are only reviewed by their optometrist 6 weeks later. While I am not advocating the pros and cons of this, one could argue there is no evidence base for or against these arbitrary time points aside from the fear of CMO or endophthalmitis. In light of this, these authors should be commended for providing an evidence base to change or direct clinical practice. Their questionnaire may be the first steps in rationalising this practice of review schedules.

However, in many other countries which are not so closely linked to insurance based systems as the USA, the ship has already sailed.”

Thank you for this comment. Indeed, practices differ significantly worldwide, and in many cases, there is no evidence behind these varied practice patterns. Even the “day 1-week 1-month 1” pattern practiced widely in the US is not based on thorough prior investigation but is mostly an anecdotal follow-up schedule. In the Preferred Practice Pattern report, the American Academy of Ophthalmology recommends a first follow-up visit within 48 hours and a second one within 1-4 weeks for refractive correction after small-incision cataract surgery (Olson et al., Ophthalmology. 2017 Feb;124(2):P1-P119). Our previous study showed that the rate of management change at the POW1 timepoint is very low (0.9%), suggesting that a POW1 visit may be safely omitted in a subset of patients who underwent routine cataract surgery (Borkar et al., Am J Ophthalmol. 2019 Mar;199:94-100). This study provides an alternative method of follow-up at POW1, which identifies patient who need management change with greater sensitivity and adds further evidence in the poorly-studied area of cataract surgery follow-up.

“This article could have the message "questionnaire means no more week 1 check ups!") so is a valid cost saving message, if the authors could specifically calculate how using this survey impacted on costs to the patient.”

Thank you for your comment. We added the suggested statement in the Conclusion (page 20, lines 361-362). A cost analysis of this follow-up pattern is beyond the scope of this study, but is planned to be the subject of interest of a future investigation, as mentioned in the Discussion (page 20, lines 354-357).

Reviewer #2: This is a well packaged message highlighting the optimal follow-up schedule after cataract surgery, nice reading prior work and how this fits in their overall work.

Thank you for your comments.

Reviewer #3: This is a very well written article that questions the current recommended followup schedule after uncomplicated cataract surgery. The study suggests a useful and simple questionnaire to be conducted via virtual visit which can help eliminate postoperative week 1 visit. The sensitivity of the study was excellent at 100%. What was the specificity of the study?

This is a great question. Although Figure 1 provides a hint to the specificity at each cutoff point, we reported only the sensitivity in the Results, because this is a screening test, and therefore false negatives must be as low as possible. In other words, we intended to sacrifice specificity in favor of sensitivity. However, this screening test demonstrated both high sensitivity and high specificity, making it an appealing tool for the risk stratification of patients following cataract surgery. We added the specificity values in the Results section (page 15, line 244).

Line 70: The sentence should read as “ Most commonly in the United states, the patients are seen postoperatively on day 1, one week, and one month after cataract surgery.

We made the change, as suggested (page 4, lines 70-72).

Line 74: Can you please be specific about couple of weeks? It can be 4-6 weeks after surgery.

The AAO recommends 1-4 weeks following small-incision cataract surgery. We specified the time interval, as suggested (page 4, line 76).

Methods: It is important to mention that this is a prospective cohort study here. Also the informed consent should be discussed at the beginning of methods rather than at the end of methods section.

We made these changes, as suggested (page 5, lines 91-103).

Line 111: The sentence can be re written as “The study population comprised of patients ….informed consent.”.

The sentence was revised as suggested (page 7, lines 127-130).

“Line 118: Figure 1 can be moved to results section. Also it is important to explain ROC figure since it is not very self explanatory. It is not clear what the numbers stand for.

Please write figure legends for all the figures at the end of the article”

Thank you for your comments. Indeed, Figure 1 belongs to the Results and was moved to the end of the manuscript, as recommended. In the ROC analysis, we sought to determine the optimal cutoff point of positive answers, in other words, what is the sensitivity of the screening test (i.e., the test’s ability to not miss patients who need management change) in patients who have at least 1 positive answer, at least 2 positive answers, at least 3 positive answers, etc. We found that sensitivity is high only for patients with at least 1 positive answer and is compromised significantly for other cutoff points. The AUC is an indicator of the screening test’s discriminative ability, in other words the test’s ability to distinguish between patients who will need management change and patients who will not. The higher the AUC, the higher the test’s discriminative ability.

---

## [Editor Report · Decision Letter 1]

5 Aug 2019

Optimization of Cataract Surgery Follow-up: A Standard Set of Questions Can Predict Unexpected Management Changes at Postoperative Week One

PONE-D-19-16669R1

Dear Dr. Kloek,

We are pleased to inform you that your manuscript has been judged scientifically suitable for publication and will be formally accepted for publication once it complies with all outstanding technical requirements.

With kind regards,

Cesario Bianchi

Academic Editor

PLOS ONE

Additional Editor Comments (optional):

Dear Dr. Kloek,

Thank you for carefully answer all questions and concerns from all parts involved (editorial and reviewer). I am glad to accept your manuscript at this time.
---

## [Editor Report · Acceptance letter]

10 Sep 2019

PONE-D-19-16669R1 

Optimization of Cataract Surgery Follow-up: A Standard Set of Questions Can Predict Unexpected Management Changes at Postoperative Week One 

Dear Dr. Kloek:

I am pleased to inform you that your manuscript has been deemed suitable for publication in PLOS ONE. Congratulations! Your manuscript is now with our production department. 

With kind regards,

on behalf of

Dr. Cesario Bianchi 

Academic Editor

PLOS ONE